# Comparative Analysis and Design of Double-Rotor Stator-Permanent-Magnet Motors with Magnetic-Differential Application for Electric Vehicles

**Tengbo Yang [1], Kwok Tong Chau [1,*], Wei Liu [1], Tze Wood Ching [1] and Libing Cao [2]**

[1]   Department of Electrical and Electronic Engineering, The University of Hong Kong, Hong Kong, China
[2]   School of Electrical and Electronic Engineering, Nanyang Technological University,
     Singapore 639798, Singapore
*    Correspondence: ktchau@eee.hku.hk

**Abstract:** In order to get rid of the bulky and lossy differential gears and to enhance the system robustness, the magnetic differential (MagD) system is proposed after the mechanical differential (MechD) and electronic differential (ElecD) systems. The MagD system is mainly composed of the double-rotor (DR) stator-permanent-magnet (PM) motor with a new set of winding whose magnetic field reversely interacts with the PM field in two rotors. As a result, the compactness and reliability of the system are improved. This paper quantitatively compares and analyzes the three major types of stator-PM motors applied in the MagD system, which can give an essential guideline on the choice of motor types in various situations. All kinds of motors are optimized in the same exercise, and their performances are thoroughly evaluated and compared by using three-dimensional finite element analysis. Finally, the motor with the best overall performance is prototyped, and the MagD system is set up for experimental verification of the optimized flux-switching PM motor.

**Keywords:** stator-permanent-magnet (PM); magnetic differential; doubly salient; flux reversal; flux switching

## 1. Introduction

As a prospective kind of mitigation of climate change and environmental deterioration, renewable energy gains its popularity in the form of the electrification of transportation [1,2]. The replacement of traditional internal combustion engine (ICE) vehicles with electric vehicles (EVs) takes up a large part in such transportation electrification [3]. Due to the upsurge of EVs, countless studies have been conducted from a wide range of perspectives. For example, authors in [4] discussed the economic impacts of the EV market. The infrastructure concerns, especially on the construction and operation of EV charging stations, were provided in [5]. Another fashion of wireless charging, namely, roadway charging, was proposed along with the consideration of energy security, and the authors presented an energy encryption strategy based on the chaos theory [6]. It is noteworthy that as a key part of EVs, significant achievements have been reached in electric motors regarding lower cost, higher power density, higher efficiency, and so on [7–9]. Among various types of motors, the stator-permanent-magnet (PM) motor, a type of motor that locates its PM materials on the stator, displays a better cooling condition, a simple rotor structure, and better robustness given its stator-fixed PM placement [10,11].

In the propulsion system of the EV, the vehicle differential system is a crucial part of directing the power from the engine to the wheels, and it is updated from time to time. The core function of a differential system is to provide different speeds for two wheels when the vehicle is cornering, thus preventing vehicle slip and the wear-out of tires. The commonly seen mechanical differential (MechD) systems, including the basic open differential, the limited-slip differential, and even the active MechDs, are trusted

for their reliability [12]. However, though devices such as onboard control processing units that receive various inputs, such as yaw rate, lateral acceleration, and steering angle, are introduced to the MechD system, making that system more controllable [13], the MechD system inevitably suffers from bulky sizes and low overall efficiency. Therefore, the electronic differential (ElecD) system, adopting multiple motors to individually drive different wheels, is proposed to get rid of the inefficient MechD system. Benefitting from the independent power and torque feeding, the ElecD system can not only avoid the wheel slip without extra transmission loss but also provide a faster response to the vehicle cornering [14]. In [15], an adaptive differential control system was proposed for four-wheel independent drive EVs. This strategy totally gets rid of the conventional differential system while distributing different torques to four wheels by four hub motors. However, the addition of extra motors increases the onboard space consumption. Additionally, controlling each wheel separately complexifies the control strategy, and serious errors may be encountered when control faults occur.

In order to combine the features of high efficiency, high reliability, and robustness and also enhance the compactness of the differential system, the magnetic differential (MagD) system is brought up to give a brand-new solution to the vehicle differential system. This MagD system tactfully installs a group of magnetic coupling (MC) windings in the motor [16]. Moreover, together with the selection of a stator-PM double-rotor (DR) axial-field (AF) motor, the MagD system can thereafter satisfy the high soundness and compactness requirements of the vehicle differential system, and the power density and overall efficiency can be boosted as well [17].

The authors presented the application of the MagD system utilizing the flux-switching PM (FSPM) motor in [17]. Nevertheless, no research on the other two common stator-PM motors in the AF-DR form, that is, doubly salient PM (DSPM) motor and flux reversal PM (FRPM) motor, was conducted to show their performances when applied to the MagD system. As these three types are the major types of stator-PM motors [18], it is necessary to find out the performance of all the aforementioned stator-PM motors when applied to the MagD system and to correctly build up the MagD system with the most suitable motor for diversified applications. By conducting three-dimensional (3D) finite element analysis (FEA), this paper aims to analyze and compare three types of AF-DR stator-PM motors with magnetic differential application in a quantitative way. A prototype with superior overall performance is manufactured to validate the analysis.

This paper is organized as follows: In Section 2, the DSPM and FRPM motors for the MagD system are proposed, and their operation principles, as well as the approach to optimize the three motors, are explained. Then in Section 3, their performances are investigated, and the quantitative comparison is illustrated in terms of the back electromotive force (EMF), torque, loss, and efficiency. After the comparison, experiments on the motor prototype are conducted, and the results are shown in Section 4. Finally, the conclusion is drawn in Section 5. The contributions of this paper can be summarized as follows:

1.  Provided a review of major differential systems in the market and discussed both their pros and cons;
2.  Proposed two novel motors for MagD systems, namely, the DSPM and FRPM motors with MC windings tactfully located in the motors;
3.  Thoroughly investigated the performances of three major types of stator-PM motors for the MagD application by using the 3D FEA simulation;
4.  Compared three motors and suggested suitable motor types on the basis of various application scenarios;
5.  Fabricated a motor prototype to conduct experimentations to validate the aforementioned theoretical analysis and simulation.

## 2. Motor Structures, Operation Principles, and Optimization

### 2.1. Topologies and Operation Principles

It should be noted that these three motors in the discussion are all functioning based on air-gap magnetic field modulation. Therefore, the stator-slot/rotor-pole number combination will hugely affect their performance [19,20]. In general, for a three-phase DSPM motor, its stator-slot/rotor-pole combination satisfies [21]:

$$P_r = Z_s \pm \frac{Z_s}{m} \tag{1}$$

where $P_r$ is the rotor-pole number, $Z_s$ is the stator-slot number, and $m$ is the phase number. Moreover, its stator-slot number $Z_s$ should follow [21]:

$$Z_s = 2mi \tag{2}$$

where $i$ is a positive integer. Besides, for the FRPM motor, its stator-slot/rotor-pole combination is governed by [22]:

$$P_w = \frac{kZ_s}{2} \pm P_r \tag{3}$$

for which $P_w$ is the pole pairs of armature windings, and $k$ is a positive odd number. Lastly, for the FSPM motor, the relationship between its winding pole pairs, rotor-pole number, and stator PM pole pairs $P_s$ observes [23]:

$$P_w = |P_s - P_r| \tag{4}$$

Further, based on the above numerical constraints, effortless investigation on the combination mentioned above has been conducted to find superior solutions, either in analytical ways or by listing the motor performance under different stator-slot/rotor-pole number combinations using FEA [24–26]. In this paper, each motor is set to have 12 stator slots for a fair and intuitive comparison. Each of them takes on a stator-slot/rotor-pole number combination that has been validated in [27–29], that is, 12 stator-slot and 8 rotor-pole (12 s/8 p) for a DSPM motor, 12 s/16 p for an FRPM motor, and 12 s/10 p for an FSPM motor. Then, the motor topologies, along with their 3D exploded views of the FSPM motor and two newly proposed motors, are shown in Figure 1a–c.

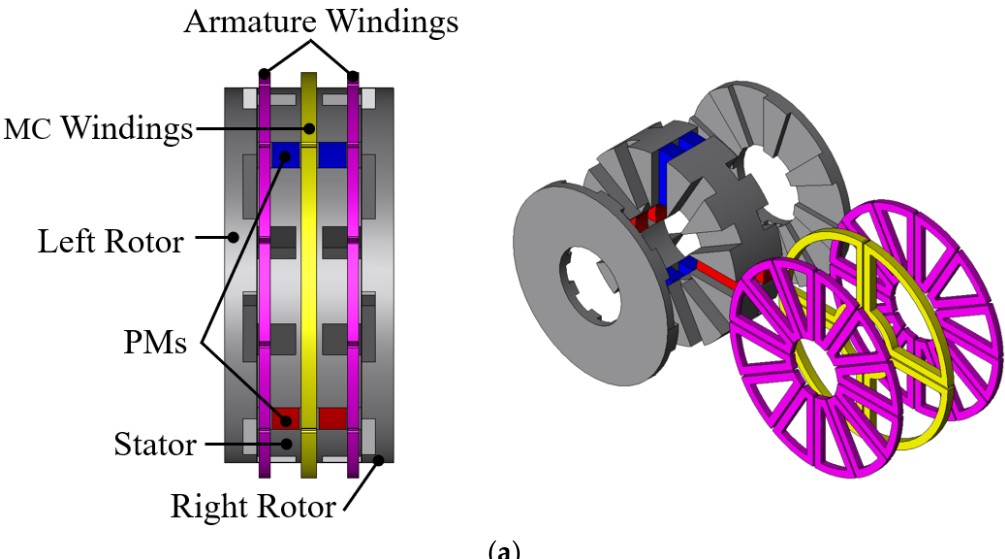

(**a**)

**Figure 1.** *Cont.*

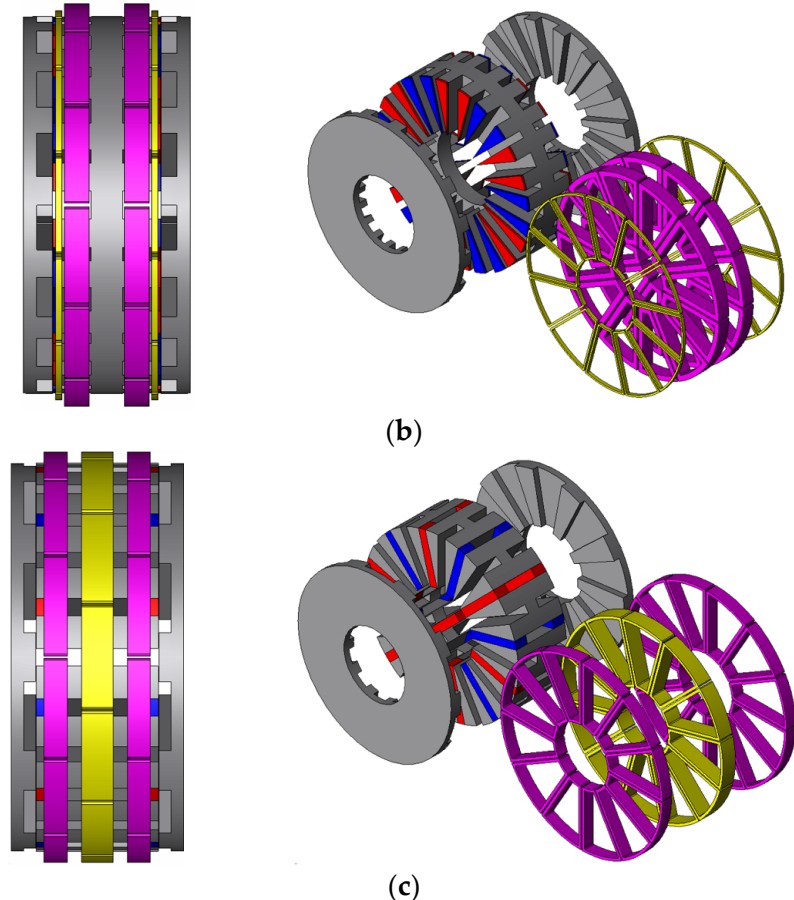

**Figure 1.** Topologies and 3D exploded views of three AF-DR motors: (**a**) DSPM, (**b**) FRPM, (**c**) FSPM.

As depicted in Figure 1, all PM materials are located in the stator, sandwiched by two simply structured iron rotors. Armature windings are placed on the two sides of the stator. MC windings are located in the middle of the stator concerning the DSPM and FSPM motors cases. On the other hand, to avoid the short circuit of the MC flux, the MC windings are split into two parts to be located on two ends of the stator in the FRPM motor.

The magnetic flux distributions with different rotor positions are depicted in Figure 2 to illustrate the operation principles of the motors. All three motors operate under a similar principle: when no cornering action is required, no current is injected into the MC windings. Thus, there is only PM flux in the motor, as shown in the distribution of purple lines in Figure 2; when the vehicle is moving curvilinearly, MC flux will be generated in the motor due to the injected DC current in the MC windings. It can be seen in Figure 2 that, on the upper rotors of the DSPM and FSPM motors and the lower rotor of the FRPM motor, the MC flux strengthens the overall flux flowing through the paths, while on the opposite rotors, the flux is weakened. Consequently, the torque produced from two sides will no longer be equal, leading to the different speeds of two rotors, then the vehicle starts cornering. As a result of cornering, two rotors will no longer be aligned. Nevertheless, the differential torque to rotors can still be created as the overall flux remains regulatable by the MC flux, as depicted in Figure 2b. Therefore, the cornering of the vehicle can be continuously and successfully achieved.

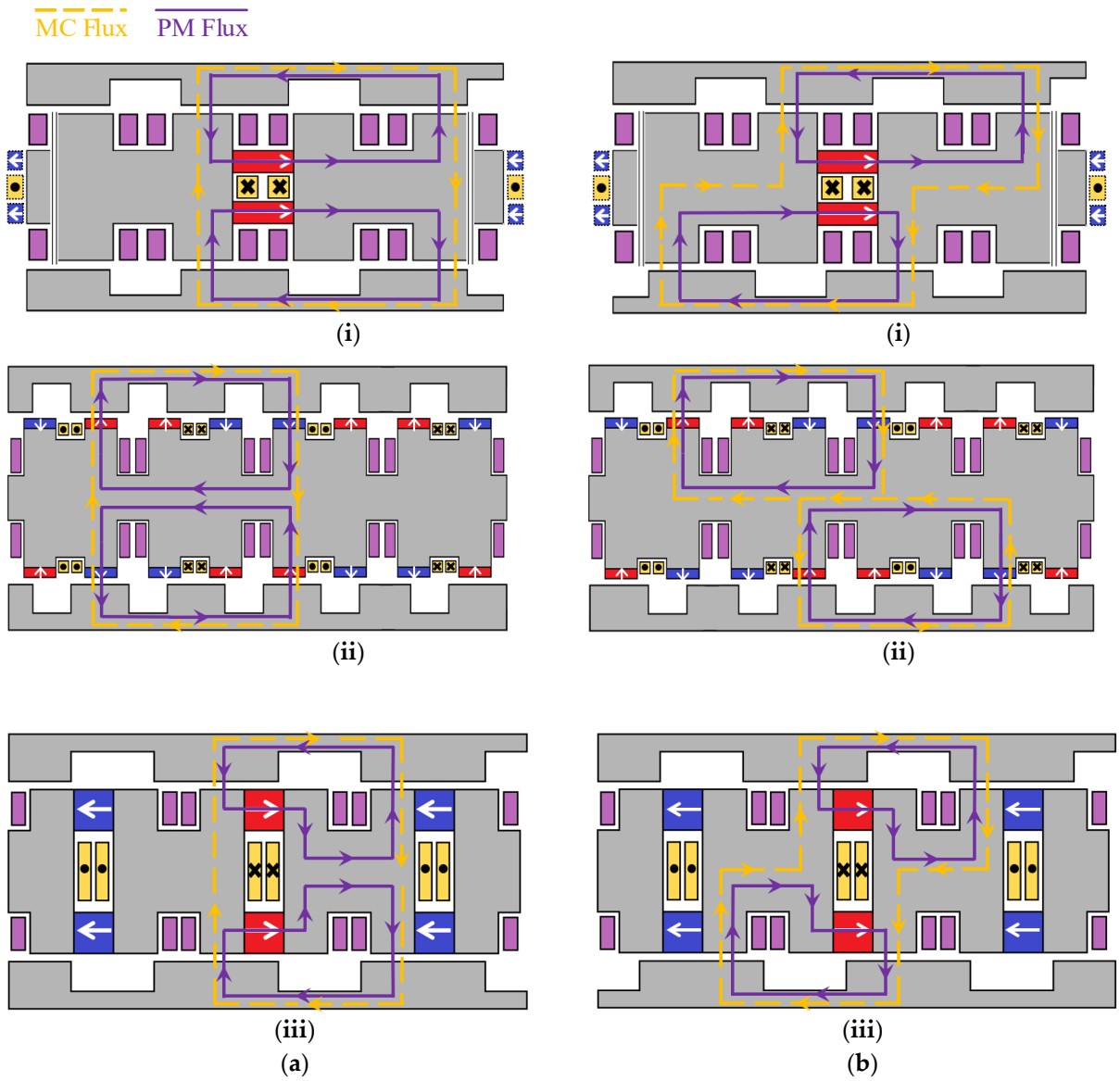

**Figure 2.** Operation principles of three AF-DR motors with rotors. (**a**) Aligned position: (**i**) DSPM, (**ii**) FRPM, (**iii**) FSPM. (**b**) Unaligned position: (**i**) DSPM, (**ii**) FRPM, (**iii**) FSPM.

## 2.2. Optimization

Figure 3 depicts the key design parameters of the proposed AF-DR FRPM motor, which for the DSPM and FSPM motors are omitted due to similarity. The slot adopts a rectangular shape to ease the complexity of motor fabrication, as mentioned in [17]. Hence, the rotor slot arc $B_{rs}$, MC winding arc $B_{mc}$, and armature winding arc $B_{ac}$ are calculated based on the centerline, as shown in Figure 3b. In the case of the DSPM motor, its MC winding arc and PM arc are both equal to the armature winding arc $B_{ac}$. The optimization of the DSPM and FRPM motor follows the same practice used for the FSPM motor in [17], aiming at providing a reasonably large torque and suppressing the torque ripple. Additionally, the analysis is conducted by 3D simulation using the FEA software JMAG. To make the comparison fair enough, all motors take on the same constraints in [17], that is, 420 mm outer diameter $D_o$ for the motor, 0.6 mm air-gap length $g$, and 186.4 mm stack length $L_{stk}$. The slot filling factor is selected as 0.6.

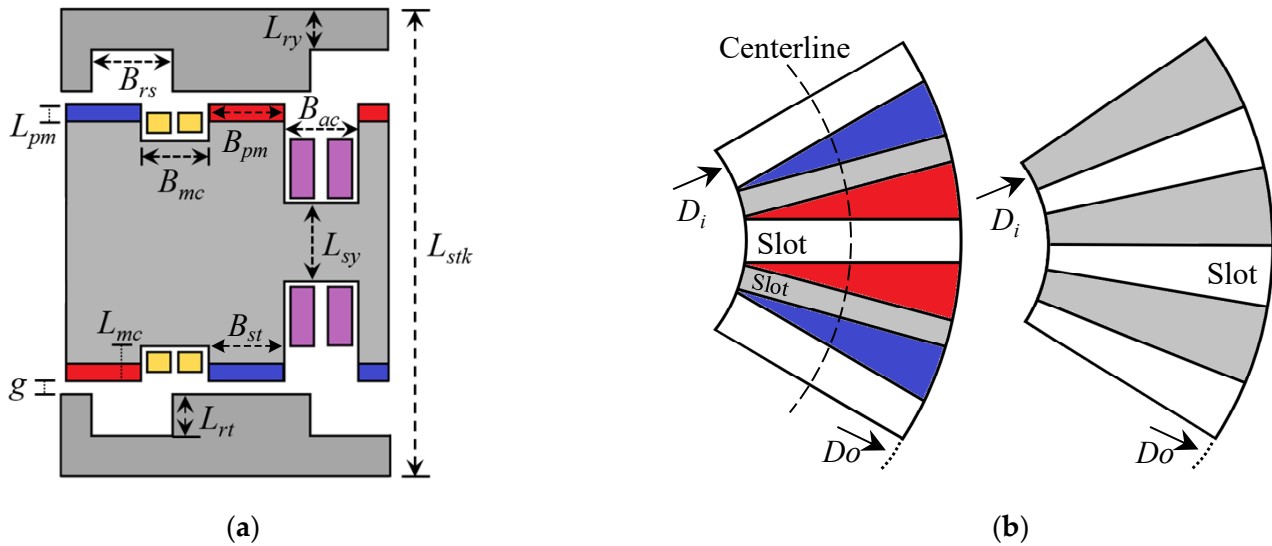

**(a)**                    **(b)**

**Figure 3.** Sketch of the AF-DR FRPM motor with key design parameters: (**a**) cylindrical view; (**b**) top view.

One of the most key optimization objectives is the torque of the motor. Initially proposed in [30], Equation (5) has a profound impact on a lot of AF motors' design [31–33]:

$$T_{em} = \frac{\pi}{4} B_{ave} A_{in} k_d k_{io} \left( 1 - k_{io}^2 \right) D_o^3 \tag{5}$$

where $T_{em}$ is the electromagnetic torque, $B_{ave}$ is the average flux density in the air gap, $A_{in}$ is the electrical loading, $k_d$ is the distribution factor, and $k_{io}$ is the ratio of inner and outer diameter or the split ratio that equals $D_i/D_o$. From Equation (5), one can see that, apart from the constraints, the split ratio $k_{io}$ is the key factor for determining the motor torque. Thus, the split ratio should be optimized priorly. Next, the PM thickness $L_{pm}$, rotor slot arc $B_{rs}$, armature winding arc $B_{ac}$, thickness of the rotor yoke and teeth $L_{ry}$ and $L_{rt}$, and stator yoke thickness $L_{sy}$ are optimized orderly, and the results are shown in Figure 4. Note that arcs are in degree and lengths are in mm.

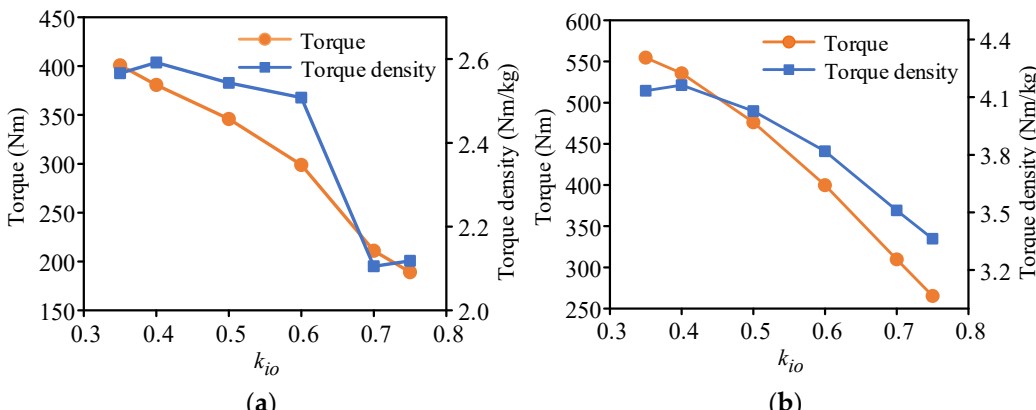

**(a)**                    **(b)**

**Figure 4.** *Cont.*

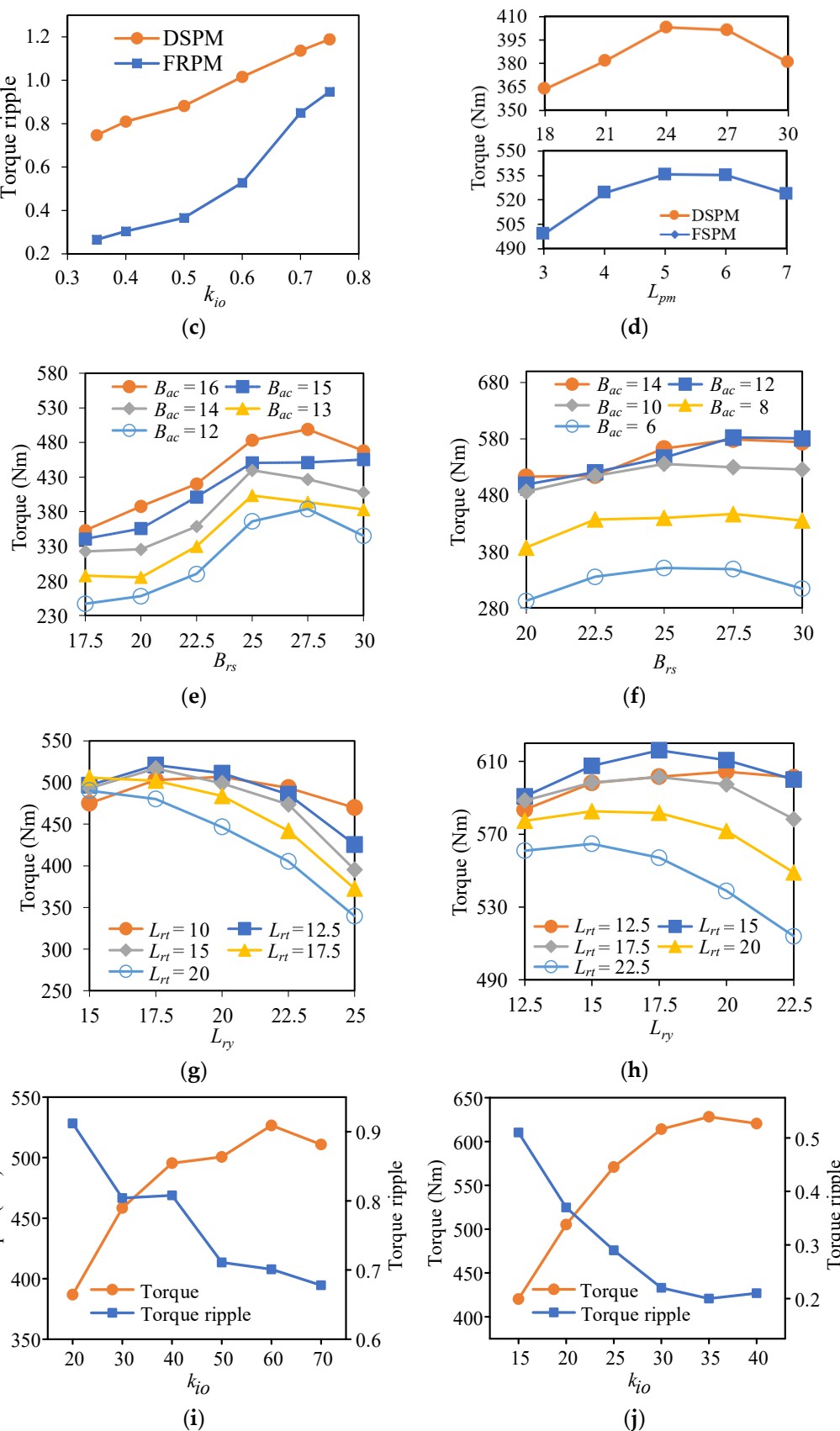

**Figure 4.** Optimization results. (**a**) $k_{io}$—DSPM, (**b**) $k_{io}$—FRPM, (**c**) torque ripple vs. $k_{io}$, (**d**) torque vs. $L_{pm}$, (**e**) $B_{rs}$ and $B_{ac}$—DSPM, (**f**) $B_{rs}$ and $B_{ac}$—FRPM, (**g**) $L_{ry}$ and $L_{rt}$—DSPM, (**h**) $L_{ry}$ and $L_{rt}$—FRPM, (**i**) $L_{sy}$—DSPM, (**j**) $L_{sy}$—FRPM.

It can be seen from Figure 4a–c that with the split ratio $k_{io}$ increasing, the torque drops while its ripple goes up. Here, the torque refers to the summation of the left and right rotor torque. Note that the smaller the $k_{io}$, the heavier the motor, since the motor volume is mainly determined by it. Thus, taking the torque density into consideration, it can be seen that $k_{io}$ = 0.4 is a reasonable option for both motors to achieve large torque while maintaining a relatively low torque ripple. Initially, the PM thickness $L_{pm}$ for the DSPM and FRPM motors are 30 and 5 mm, respectively. According to Figure 4d, given that the PM materials are expensive, they are optimized to be 24 and 5 mm after optimization.

Two arc parameters, namely, the armature winding arc $B_{ac}$ and the rotor slot arc $B_{rs}$, play important roles in the determination of motor torque performance. For the proposed DSPM motor, its constraints on the stator arc parameters are:

$$\begin{cases} B_{ac} = B_{pm} = B_{mc} \\ 2B_{ac} + 2B_{st} = \frac{\pi}{6} \end{cases} \tag{6}$$

where $B_{pm}$, $B_{mc}$, and $B_{st}$ are the PM arc, MC winding arc, and stator teeth arc, respectively. When $B_{ac}$ increases, the current flowing through the winding slot will increase, and so does the space to accommodate PM materials. Thus, the torque performance will be mainly restrained by saturation of stator and rotor yokes. From Figure 4e, it can be found that the torque increases together with $B_{ac}$. However, as regards the proposed FRPM motor, its stator arc parameters are governed by:

$$\begin{cases} B_{ac} + 2B_{pm} + B_{mc} = \frac{\pi}{6} \\ B_{pm} = B_{st} \end{cases} \tag{7}$$

As a result, the increase in $B_{ac}$ will take up more space for locating PM materials and accelerate the saturation of the stator. In this case, there exists a $B_{ac}$ to obtain the optimal torque output, as shown in Figure 4f. In terms of the rotor slot arc, one can see from Figure 4e,f that the torque increases at the beginning and then decreases with the growth of $B_{rs}$. Consequently, $B_{ac}$ and $B_{rs}$ are optimized as 16° and 27.5° for the DSPM motor and 12° and 27.5° for the FRPM motor.

The other two parameters, rotor yoke length $L_{ry}$ and rotor teeth length $L_{rt}$, which significantly influence the armature winding area, stator volume, and saturation in both the stator and rotor, are also taken into consideration in the optimization. As shown in Figure 4g,h, with the increase in both $L_{ry}$ and $L_{rt}$, the torque rises first and then drops. This is because the increase in the stack length of the rotor will reduce the armature winding area, which limits the input power. Additionally, the stator is more likely to saturate due to its shrinking size. As a result, the optimal $L_{ry}$ is 17.5 mm for both motors. Further, $L_{rt}$ is optimized as 12.5 mm for the DSPM motor and 15 mm for the FRPM motor.

Finally, the thickness of the stator yoke is optimized. As depicted in Figure 4i,j, the torque initially ascends and then goes down, while the torque ripple behaves reversely. The reason lies in the ease of saturation due to the $L_{sy}$ increment. According to the optimization results, the optimized design parameters of the proposed DSPM and FRPM motors are listed in Table 1, while that of the FSPM motor will be adopted from [17]. Note that the height of MC winding, $L_{mc}$, is kept constant during the optimization.

**Table 1.** Design specifications and parameters.

| Parameters | DSPM | FRPM | Parameters | DSPM | FRPM |
|---|---|---|---|---|---|
| Outer diameter | 420 mm | | Rotor slot arc | 27.5° | |
| Inner diameter | 168 mm | | AC winding arc | 16° | 12° |
| Air-gap length | 0.6 mm | | MC winding arc | 16° | 6° |
| Axial stack length | 186.4 mm | | Turns per AC coils | 20 | |
| Rotor yoke length | 17.5 mm | | Turns per MC coils | 30 | |
| Rotor teeth length | 12.5 mm | 15 mm | Slot filling factor | 0.6 | |
| PM thickness | 24 mm | 5 mm | PM material | N35SH | |
| Stator yoke thickness | 60 mm | 35 mm | SMC material | Somaloy 700-3P | |
| Height of MC winding | 22 mm | 20 mm | No. of rotor slots | 8 | 16 |

## 3. Comparison of Motor Performance

By conducting the 3D FEA in the JMAG software, the electromagnetic performances, including no-load back-EMF, electromagnetic torque, differential torque and efficiency, and iron loss of all three motors, are investigated and compared. Considering a scenario under the natural cooling condition, the armature current density is selected as 6 A/mm$^2$, and the rotor speed is chosen to be 900 rpm. The simulation results are presented in the following.

### 3.1. No-Load EMF

Figure 5 illustrates the A-phase no-load back EMFs of three motors. It can be seen that the FSPM motor shows a more sinusoidal waveform and a significantly larger amplitude than its two counterparts due to its exquisite structure. On the contrary, the DSPM and FRPM motors show trapezoidal waveforms. Therefore, the FSPM motor adopts the brushless (BL) AC control strategy, whereas BLDC controls for the other two motors.

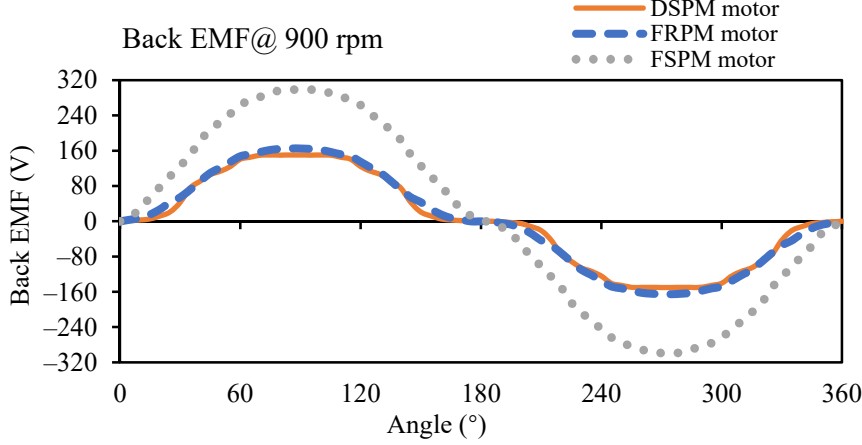

**Figure 5.** Back-EMF waveforms of three AF-DR motors.

### 3.2. Electromagnetic Torque

Simulated with the same 6 A/mm$^2$ current density $J_{ac}$, the torque performances of the motors are shown in Figure 6a. Similar to the traditional comparison of these three motors, the FSPM motor shows the best torque performance, that is, the highest torque and lowest torque ripple, while the DSPM motor is inferior to the other two counterparts. Additionally, the average torque versus $J_{ac}$ is evaluated as depicted in Figure 6b, where all motors saturate more or less when the current density increases.

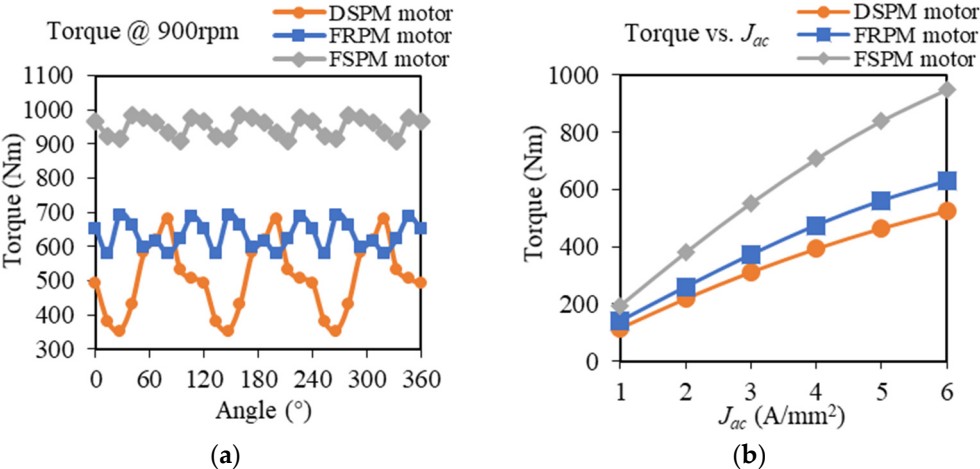

**Figure 6.** Torque performances of three AF-DR motors without MC current: (**a**) $J_{ac} = 6$ A/mm²; (**b**) different armature current densities.

### 3.3. Differential Torque

In the cornering situation, which means that the MC current is injected, it can be known from Figure 7 that all motors can create differential torques because of the MC flux regulation. Both the torques of the DSPM and FSPM motor become saturated with the increase in the MC current, while for the FSPM motor, the torques on its two rotors change linearly in the given range, meaning that the vehicle cornering has no side effect of saturation to the motor. This feature guarantees that the total torque provided by the motor can remain stable during cornering, which can be seen from Figure 7d. It should be noted that keeping torque output in a stable state is of vital essence when the near-maximum torque output is constantly needed by vehicles, for example, overtaking on freeways and climbing on steep slopes.

### 3.4. Efficiency and Loss

As listed in Table 1, the material for the stator and rotor adopts the soft magnetic composite (SMC) for the motors' axial-field structure. This material has noticeable advantages over conventional solid silicon steel, such as being easy to construct and suppressing the eddy current.

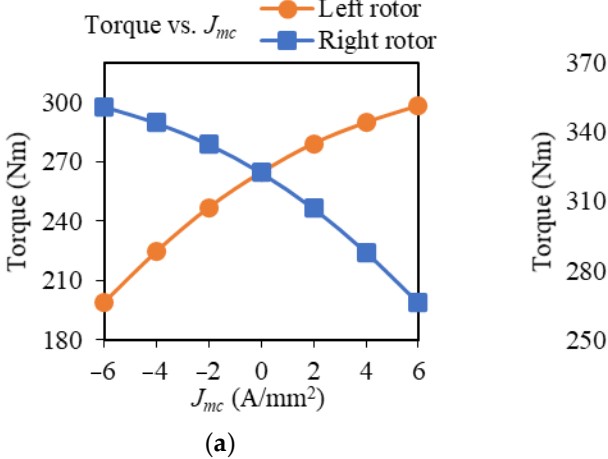

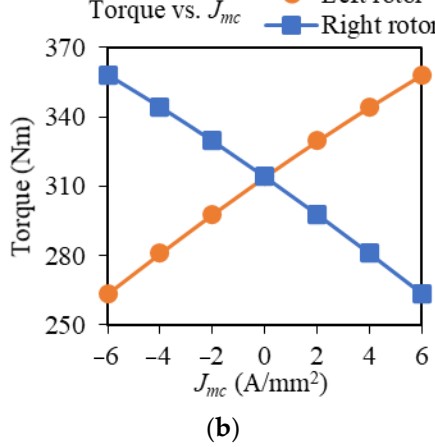

**Figure 7.** *Cont.*

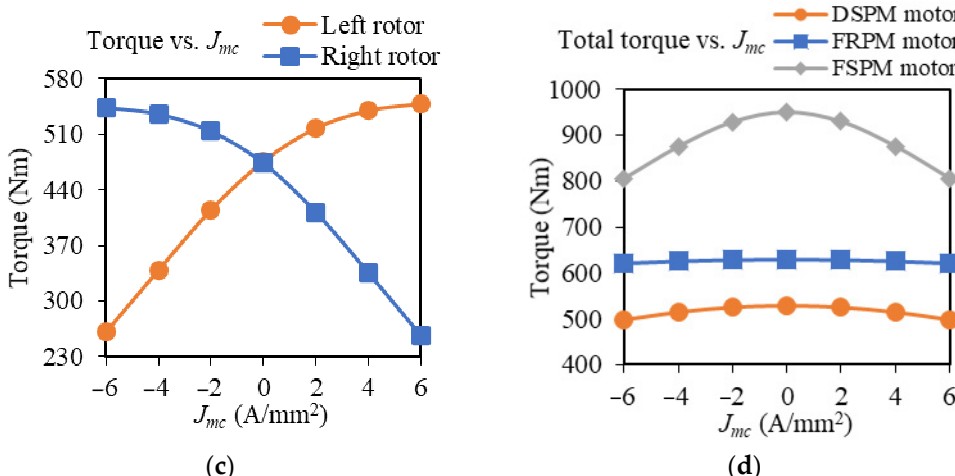

**Figure 7.** Torque performances of three AF-DR motors under different MC current densities: (**a**) DSPM, (**b**) FRPM, (**c**) FSPM, (**d**) total torque.

Figure 8a illustrates that with the injection of the MC current, extra copper loss is introduced; thus, all motors experience efficiency drops. Besides, due to the reduction of output torque with the increase in MC current, the efficiency of the DSPM motor and FSPM motor is further decreased. As regards the iron loss, the FRPM motor shows the largest iron loss due to its 12 s/16 p structure, which means that with the same rotor speed, the FRPM motor applies the highest armature current frequency. Hence, its iron loss is remarkably larger in comparison.

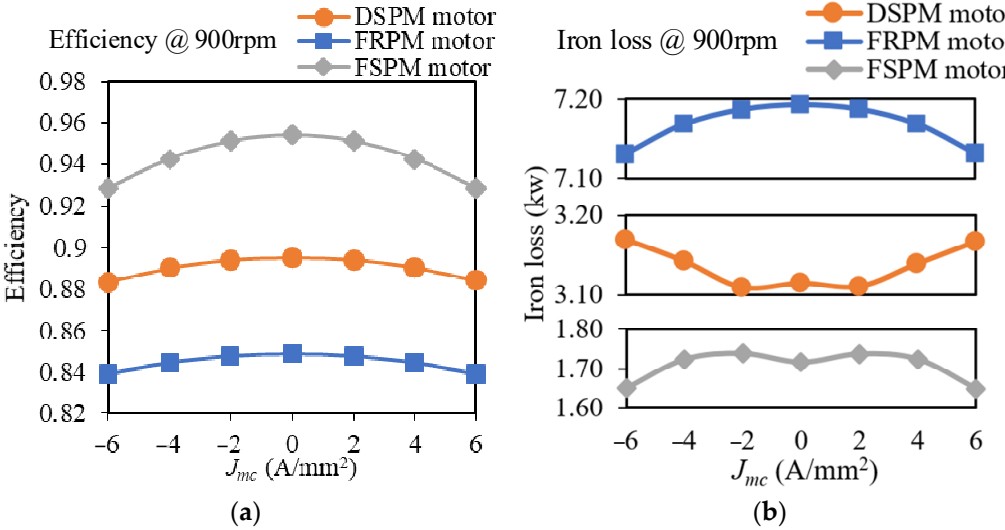

**Figure 8.** Efficiency and iron loss of three AF-DR motors under different MC current densities: (**a**) efficiency; (**b**) iron loss.

The performance comparison of the three motors is summarized in Table 2. It shows that in terms of the stator, rotors, and PMs in total, the FRPM motor features the smallest volume with the same 0.4 split ratios. Especially, the least PM materials consumption is achieved on the FRPM motor over three optimized motors. As a result, for the torque per PM volume, the FRPM motor excels 3.5 and 2.5 times that of the DSPM and FSPM motors. As regards cost-sensitive scenarios, this advantage is of great appeal. In normal applications, one can see that the FSPM motor nearly dominates all the performance. That is, it has the largest back-EMF, which is nearly 2 times of the other motors; the largest torque and torque density; the highest efficiency plus the highest power density; and the least torque ripple and loss. On the contrary, the DSPM motor does not show any eye-catching

performance data in this comparison. Nevertheless, due to the flux interaction on two rotors of the FSPM motor, it needs an extra reverse gear to avoid the traction force between two rotors [17] when on load. It should be noted that all the comparison is based on the motors themselves, excluding any other transmission parts.

**Table 2.** Motor performance.

| Characteristics | DSPM | FRPM | FSPM |
|---|---|---|---|
| SMC volume (L) | 14.38 | 12.65 | 12.90 |
| PM volume (L) | 1.37 | 0.47 | 1.75 |
| Frequency (Hz) | 120 | 240 | 150 |
| Back-EMF amplitude (V) | 150.2 | 165.3 | 298.7 |
| Current (rms, A) | 73.5 | 89.8 | 89.4 |
| Total torque (Nm) | 526.6 | 628.2 | 950.1 |
| Torque ripple | 62.3% | 18.2% | 7.8% |
| Total torque/PM volume (Nm/L) | 384.4 | 1336.6 | 542.9 |
| Torque density (Nm/kg) | 3.58 | 4.51 | 6.04 |
| Output power (kW) | 49.62 | 59.21 | 89.54 |
| Loss (kW) | 5.81 | 10.57 | 4.29 |
| Efficiency | 89.5% | 84.9% | 95.4% |
| Power density (kW/kg) | 0.34 | 0.43 | 0.57 |

## 4. Experimental Results

Consequently, due to the outstanding performance of the FSPM motor among the three AF-DR motors, as expected, the FSPM motor is then chosen to be prototyped to obtain the experimental results. Although in the simulation above, the FSPM motor is designed to have a power level of nearly 80 kW to fit the EV propulsion system, as in reality, a downscaling prototype is manufactured in this paper to verify the analysis on purpose to ease the experiment difficulty. The SMC material is applied consistently with the simulation in this experimental prototype to suppress the eddy-current loss and simplify the manufacturing process.

Figure 9 shows the testbed setup. In addition, the key design parameters of the fabricated prototype are listed in Table 3. As shown in Figure 9, the prototype is located at the center of the testbed. Symmetrically, through an optoelectronic angular encoder and a torque sensor, the left or right side of the prototyped motor is connected to a three-phase PM synchronous motor (PMSM). These two PMSMs will drive the AF-DR FSPM motor. An oscilloscope is used for data capture, and a DC power source is installed for a flux regulation test. The no-load back-EMF of the FSPM motor is investigated in the experiment. The measured (MEA) no-load back-EMFs on both rotors under a rotating speed of 500 rpm are shown in Figure 10, with the FEA simulation results presented as well.

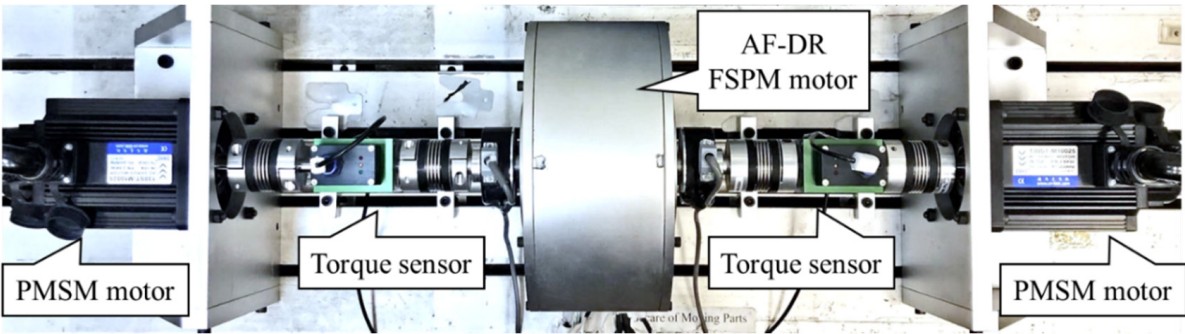

**Figure 9.** Experiment setup of the testbed.

**Table 3.** Design parameters of the prototype.

| Parameters | Value | Parameters | Value |
|---|---|---|---|
| Outer diameter | 220 mm | Turns per AC coils | 60 |
| Inner diameter | 128 mm | Turns per MC coils | 64 |
| Air-gap length | 1.0 mm | Slot filling factor | 0.4 |
| Axial stack length | 88 mm | PM remanence | 1.14 T |
| Stator poles no. | 12 | PM volume | 163 cm$^3$ |
| Rotor poles no. | 10 | SMC materials | Somaloy 700-3P |
| Coil type | AWG 22 | Outer frame material | Aluminum |

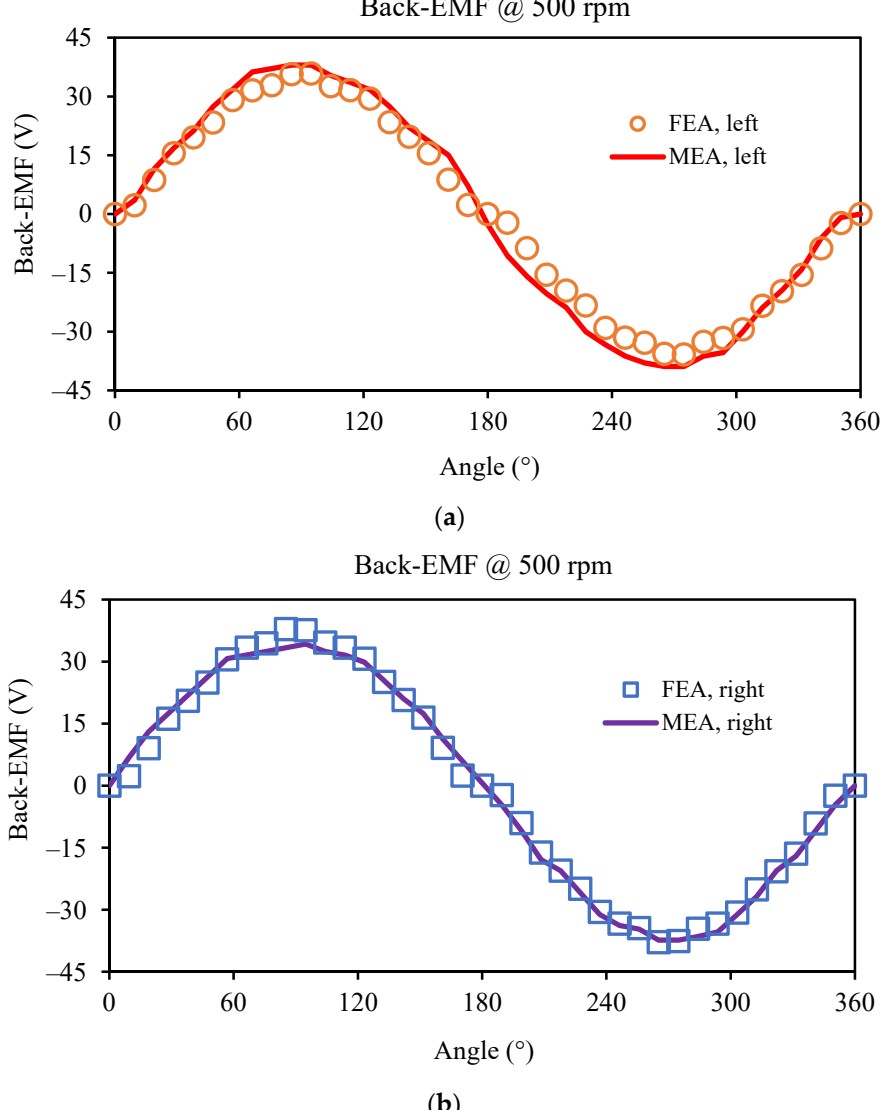

**Figure 10.** No-load back-EMFs without MC current: (**a**) left rotor; (**b**) right rotor.

From Figure 10, one can see that due to the symmetrical structure of the two rotors, the FEA results of back-EMFs on both rotors share similar patterns, having a peak value of around 35 V. For the measured amplitude of the back-EMF on the right rotor, it turns out to be nearly 34 V, which is a little lower than an amplitude of approximately 37 V on the left rotor. This is resulted from the fabrication tolerance, for example, unequal air-gap length. Thus, it can be inferred that the stator is installed slightly towards the left rotor. However, even with the inevitable fabrication tolerance, the measured results fit the FEA results well. Then, the MC current is injected into the MC windings to validate their ability

to regulate the air-gap flux density reversely on two rotors. The results are illustrated in Figure 11. It can be seen that while increasing the MC current from −4 to 4 A, the back-EMF amplitude on the right rotor decreases from 50 V to less than half of it, around 24 V in the FEA condition. Meanwhile, for the measured back-EMF, it drops from nearly 45 V to a similar endpoint. Nevertheless, on the other hand, the back-EMF amplitude on the left rotor is lifted along the same MC current change for both FEA and MEA conditions. Besides, the phenomenon resulting from the unequal air-gap length can also be seen in Figure 11. The balanced point of the experimental result shifts slightly from the theoretical point when the MC current is equal to zero. However, overall, the MEA results meet well with the FEA results, and the amplitudes of back-EMF on two rotors are proven to be successfully reversely regulated by the MC current in the experiment.

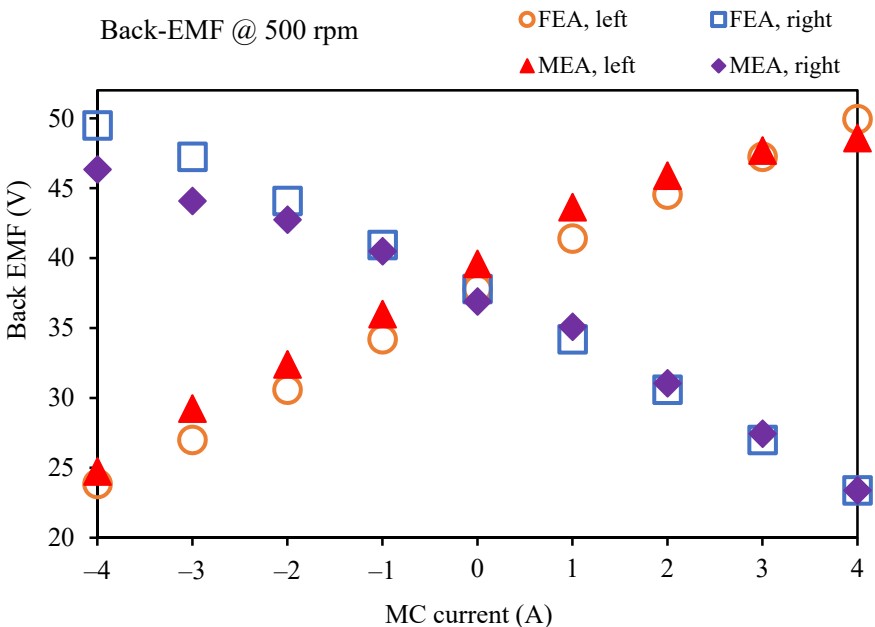

**Figure 11.** No-load back-EMFs with MC current on two rotors.

## 5. Conclusions

This paper presented a thoroughly quantitative comparison and analysis of three types of AF-DR stator-PM motors with magnetic differential for EVs by using 3D FEA. When the MC windings are added to the motor to reversely regulate the magnetic flux on two sides of the motors, all three kinds of motors can successfully function as a differential. The optimizations of the DSPM and FRPM motors are given in detail, aiming at the relatively high torque and reasonably low torque ripple. The system performances of three motors are simulated and assessed, including the no-load back-EMF, the static torque, and the differential torque, as well as their efficiency and loss. In this study, it can be found that the DSPM motor holds the merit of the simplicity of construction while showing a mediocre performance among all three types of motors. The FRPM motor is attention worthy for the relatively high production of torque per PM consumption, which, industrially speaking, is attractive to cost-sensitive cases. Moreover, its ability to maintain the output torque when the vehicle is in the curvilinear movement guarantees the safety of the driving. Nevertheless, the FRPM motor may suffer from the most significant iron loss because it applies the highest frequency when rotating at the same speed as its counterparts. Within the given sizing constraints, the optimized FSPM motor provides the largest torque and output power, as well as the lowest torque ripple and iron loss. Therefore, the highest efficiency and power density are obtained with the FSPM motor when ignoring any other transmission part, thus making it a competitive candidate for the MagD system. Finally, given the excellent simulation results of the FSPM motor, a downscaling prototype of the FSPM motor is fabricated, simulated, and experimented to show the concept of MC flux

regulation. Theoretical analysis, simulation results, and experimental results verify the feasibility and effectiveness of the optimized FSPM motor.

**Author Contributions:** Conceptualization, T.Y., K.T.C. and L.C.; methodology, T.Y. and W.L.; software, T.Y. and W.L.; validation, T.Y., W.L. and T.W.C.; formal analysis, T.Y. and L.C.; investigation, T.Y. and W.L.; resources, K.T.C. and T.W.C.; writing—original draft preparation, T.Y. and W.L.; writing— review and editing, K.T.C., W.L. and T.W.C.; supervision, K.T.C. and T.W.C.; project administration, K.T.C.; funding acquisition, K.T.C. All authors have read and agreed to the published version of the manuscript.

**Funding:** This work was supported by a grant (Project No. 17205518) from the Research Grants Council, Hong Kong Special Administrative Region, China.

**Data Availability Statement:** Not applicable.

**Conflicts of Interest:** The authors declare no conflict of interest.

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
