# Peer review of "Comparative Analysis and Design of Double-Rotor Stator-Permanent-Magnet Motors with Magnetic-Differential Application for Electric Vehicles"

_wevj, doi:10.3390/wevj13110199_

Round 1

Reviewer 1 Report

1. Concept is good but authors need to explain more indetail about the presented work.

2. Introduction must be improved with latest technologies in EV.

Mopidevi, S., Narasipuram, R.P., Aemalla, S.R. and Rajan, H. ‘E-mobility: impacts and analysis of future transportation electrification market in economic, renewable energy and infrastructure perspective’, Int. J. Powertrains, Vol. 11, Nos. 2/3, pp.264–284, 2022.

Rajanand Patnaik Narasipuram, Subbarao Mopidevi, A technological overview & design considerations for developing electric vehicle charging stations, Journal of Energy Storage, Volume 43, 2021, 103225.

3. Add research contributions point by point.

4. Results and analysis section is ok.

5. Conclusion can be improved with more acheviments.

Author Response

Reviewer: 1

General comments:

Concept is good but authors need to explain more indetail about the presented work.

Response: Thank you very much for your careful review and constructive comments. All your suggestions have been addressed in this revised version and all changes are written in red color.

Comment 1:  Introduction must be improved with latest technologies in EV.

Mopidevi, S., Narasipuram, R.P., Aemalla, S.R. and Rajan, H. ‘E-mobility: impacts and analysis of future transportation electrification market in economic, renewable energy and infrastructure perspective’, Int. J. Powertrains, Vol. 11, Nos. 2/3, pp.264–284, 2022.

Rajanand Patnaik Narasipuram, Subbarao Mopidevi, A technological overview & design considerations for developing electric vehicle charging stations, Journal of Energy Storage, Volume 43, 2021, 103225.

Response 1:   Thank you very much for your constructive suggestion. As suggested, the following discussion about the latest technologies relevant to EVs has been added in the first paragraph of Section 1 (Introduction). The change is in Line 30-35, Page 1:

“Due to the upsurge of EVs, countless research has been done from a wide range of perspectives. For example, authors in [4] discussed the economic impacts from the EV market. The infrastructure concerns especially the construction and operation of EV charging stations was provided in [5]. Another fashion of charging, the roadway charging, was proposed along with the consideration of energy security, and the authors presented an energy encryption strategy based on the chaos theory [6].

Accordingly, References have been updated by adding three references for your convenient review:

[4] Mopidevi, S.; Narasipuram, R.P.; Aemalla, S.R.; Rajan, H. E-mobility: impacts and analysis of future transportation electrification market in economic, renewable energy and infrastructure perspective. Int. J. Powertrains, 2022, 11, 264-284.

[5] Narasipuram, R. P.; Mopidevi, S. A technological overview & design considerations for developing electric vehicle charging stations, J. Energy Storage, 2021, 143, 103225.

[6] Liu, W.; Chau, K.T.; Lam, W.H.; Zhang, Z. Continuously variable-frequency energy-encrypted wireless power transfer. Energies, 201912, 1286.

Comment 2:  Add research contributions point by point.

Response 2:   Thank you very much for your careful review and constructive comments. The research contributions have been listed for clarification. The change is in Line 86-96, Page 2, Section 1:

The contributions of this paper can be summarized as follows:

  1. Provided a review on major differential systems in the market and discussed both their pros and cons;
  2. Proposed two novel motors for MagD systems, namely the DSPM and FRPM motors with MC windings tactfully located in the motors;
  3. Thoroughly investigated the performances of three major types of stator-PM motors for the MagD application by using the 3D FEA simulation;
  4. Compared three motors and suggested suitable motor types on the basis of application scenarios;
  5. Fabricated a motor prototype to conduct experimentations to validate the aforementioned theoretical analysis and simulation.

Comment 3:  Results and analysis section is ok.

Response 3:   Thank you very much for your constructive suggestion.

Comment 4:  Conclusion can be improved with more acheviments.

Response 4:   Thank you very much for your constructive suggestion. The conclusion summarizes the work done in this paper. As hinted by the simulation results, the authors provide ideas on the suitable scenarios for all three motors. As suggested, the following parts providing more details of the work done in the paper have been added for clearer explanation. The changes are in Line 371-375 and Line 384-386, Page 14, Section 5 (Conclusion):

The optimizations of the DSPM and FRPM motors are given in detail aiming at the relatively high torque and reasonably low torque ripple. The system performances of three motors are simulated and assessed, including the no-load back-EMF, the static torque, and the differential torque as well as their efficiency and loss. In this study, it can be found that the DSPM motor holds the merit of the simplicity of construction whilst showing a mediocre performance among all three types of motors.”

“Therefore, the highest efficiency and power density are obtained with the FSPM motor when ignoring any other transmission part, thus making it a competitive candidate for the MagD system. Finally, given the excellent simulation results of the FSPM motor, a downscaling prototype of the FSPM motor is fabricated, …”

Reviewer 2 Report

The present paper concerns a Comparative Analysis and Design of double-rotor (DR) stator permanent-magnet (PM) motor with Magnetic-Differential Application for EV. Three major types of stator-PM motors applied in the MagD system are compared and analyzed (DSPM,FRPM and FSPM). Then, the motors are optimized and their performances are evaluated and compared with finite element analysis.  Finally, the FSPM is prototyped, and the MagD system is set up for experimental verification of the optimized flux-switching PM motor.

This paper is technically sound in many recent references and research. The problem sounds interesting, and the proposed approach is attractive to the EV application.

- The literature review should be added and introduction should be improved, Authors should have provided a better discussion of the cited references.

- the paper missed a section that describe the adopted approach

- where is the relation between this work and the VE application in this paper

- More explanation about experiment setup of testbed

- Conclusions should contain the perspective of this work.

Author Response

Reviewer: 2

General comments:

The present paper concerns a Comparative Analysis and Design of double-rotor (DR) stator permanent-magnet (PM) motor with Magnetic-Differential Application for EV. Three major types of stator-PM motors applied in the MagD system are compared and analyzed (DSPM,FRPM and FSPM). Then, the motors are optimized and their performances are evaluated and compared with finite element analysis.  Finally, the FSPM is prototyped, and the MagD system is set up for experimental verification of the optimized flux-switching PM motor.

This paper is technically sound in many recent references and research. The problem sounds interesting, and the proposed approach is attractive to the EV application.

Response:  Thank you very much for your careful review and constructive comments. All your suggestions have been addressed and all changes are written in red color.      

Comment 1:  The literature review should be added and introduction should be improved. Authors should have provided a better discussion of the cited references.

Response 1:   Thank you very much for your constructive suggestion. We have noticed your concern about the literature review and Section 1 Introduction. The literature review has been newly included in Introduction, and many statements from Introduction have been newly summarized from the literature review of cited papers. As suggested, the following part has been added for reviewing the latest EV technologies. The change is in Line 30-37, Page 1, Section 1:

“Due to the upsurge of EVs, countless research has been done from a wide range of perspectives. For example, authors in [4] discussed the economic impacts from the EV market. The infrastructure concerns, especially on the construction and operation of EV charging stations, were provided in [5]. Another fashion of wireless charging, namely roadway charging, was proposed along with the consideration of energy security, and the authors presented an energy encryption strategy based on the chaos theory [6]. Noteworthy, as a key part of EVs, significant achievements have been reached in electric motors regarding lower cost, higher power density, and higher efficiency, etc. [7-9].”

Accordingly, References have been enriched by adding three relevant references. These references are listed here for your convenient review:

[4] Mopidevi, S.; Narasipuram, R.P.; Aemalla, S.R.; Rajan, H. E-mobility: impacts and analysis of future transportation electrification market in economic, renewable energy and infrastructure perspective. Int. J. Powertrains, 2022, 11, 264-284.

[5] Narasipuram, R. P.; Mopidevi, S. A technological overview & design considerations for developing electric vehicle charging stations, J. Energy Storage, 2021, 143, 103225.

[6] Liu, W.; Chau, K.T.; Lam, W.H.; Zhang, Z. Continuously variable-frequency energy-encrypted wireless power transfer. Energies, 201912, 1286.

Another literature review on the differential systems has been added. The changes are in Line 43-44, Page 1, and Line 47-49, Line 55-57, Page 2, Section 1:

“In the propulsion system of the EV, the vehicle differential system is a crucial part of directing the power from the engine to the wheels, and it gets updated from time to time. The core function of a differential system is to provide different speeds for two wheels when the vehicle is cornering, thus preventing the vehicle slip and the wear-out of tires. The commonly seen mechanical differential (MechD) systems, including the basic open differential, the limited-slip differential, and even the active MechDs, are trusted for their reliability [12]. However, though devices like onboard control processing units that receive various inputs like yaw rate, lateral acceleration, and steering angle are introduced to the MechD system, making that system more controllable [13], the MechD system inevitably suffers from bulky sizes and low overall efficiency. Therefore, the electronic differential (ElecD) system, adopting multiple motors to individually drive different wheels, is proposed to get rid of the inefficient MechD system. Benefitting from the independent power and torque feeding, the ElecD system can not only avoid the wheel slip without extra transmission loss but also provide a faster response to the vehicle cornering [14]. In [15], an adaptive differential control system was proposed for four-wheel independent drive EVs. This strategy totally gets rid of the conventional differential system while distributing different torques to four wheels by four hub motors.”

To further improve the Introduction, the contributions of this paper have been summarized. The change is in Line 86-96, Page 2, Section 1:

The contributions of this paper can be summarized as follows:

  1. Provided a review on major differential systems in the market and discussed both their pros and cons;
  2. Proposed two novel motors for MagD systems, namely the DSPM and FRPM motors with MC windings tactfully located in the motors;
  3. Thoroughly investigated the performances of three major types of stator-PM motors for the MagD application by using the 3D FEA simulation;
  4. Compared three motors and suggested suitable motor types on the basis of various application scenarios;
  5. Fabricated a motor prototype to conduct experimentations to validate the aforementioned theoretical analysis and simulation.

Comment 2:  The paper missed a section that describe the adopted approach.

Response 2:   Thank you very much for your constructive suggestion. We have noticed your concern about the adopted approach in this paper. As stated in Line 76-77, Page 2, Section 1, this paper adopted three-dimensional (3D) finite element analysis (FEA), and the used software is JMAG, which is shown in Line 170, Page 5, Section 2.2. The FEA method is a popular way for numerically solving differential equations. Using the FEA software can provide reasonable simulation results with high accuracy. More information about FEA can be referred to:

[R1] Bhavikatti, S. S. Finite element analysis. New Age International, 2005.

[R2] Hughes, Thomas JR. The finite element method: linear static and dynamic finite element analysis. Courier Corporation, 2012.

Comment 3:  Where is the relation between this work and the VE application in this paper?

Response 3:   Thank you very much for your constructive suggestion. We have noticed your concern about the relationship between this paper and the EV application. This paper proposed two motors that are suitable for MagD system applications. All three motors discussed in this paper serve as both vehicle motors and differentials, which are core parts of the EV propulsion systems. As stated in this paper, “it is necessary to find out the performance of all aforementioned stator-PM motors when applied to the MagD system and to correctly build up the MagD system with the most suitable motor for diversified applications”, the purpose of this paper is to find the motor with excellent performance among its counterparts. To alleviate the confusion, we have provided a summary of contributions at the end of Section 1. The change is in Line 86-96, Page 2, Section 1:

The contributions of this paper can be summarized as follows:

  1. Provided a review on major differential systems in the market and discussed both their pros and cons;
  2. Proposed two novel motors for MagD systems, namely the DSPM and FRPM motors with MC windings tactfully located in the motors;
  3. Thoroughly investigated the performances of three major types of stator-PM motors for the MagD application by using the 3D FEA simulation;
  4. Compared three motors and suggested suitable motor types on the basis of various application scenarios;
  5. Fabricated a motor prototype to conduct experimentations to validate the aforementioned theoretical analysis and simulation.

Comment 4:  More explanation about experiment setup of testbed.

Response 4:   Thank you very much for your constructive suggestion. We have noticed your concern about the explanation of the testbed setup. To better improve the understanding of the testbed, the following change has been made in Line 330-335, Page 12, Section 4 (Experimental Results):

“As shown in Figure 9, the prototype is located at the center of the testbed. Symmetrically, through an optoelectronic angular encoder and a torque sensor, the left or right side of the prototyped motor is connected to a three-phase PM synchronous motor (PMSM). These two PMSMs will drive the AF-DR FSPM motor. An oscilloscope is used for data capture and a DC power source is installed for flux regulation test.

Comment 5:  Conclusions should contain the perspective of this work.

Response 5:   Thank you very much for your constructive suggestion. The conclusion summarizes the work done in this paper. As hinted by the simulation results, the authors provide perspectives on the suitable scenarios for all three motors. As suggested, the following parts providing more details of the work done in the paper have been added for explanation. The changes are in  Line 371-375 and Line 384-386, Page 14, Section 5 (Conclusion):

The optimizations of the DSPM and FRPM motors are given in detail aiming at the relatively high torque and reasonably low torque ripple. The system performances of three motors are simulated and assessed, including the no-load back-EMF, the static torque, and the differential torque as well as their efficiency and loss. In this study, it can be found that the DSPM motor holds the merit of the simplicity of construction whilst showing a mediocre performance among all three types of motors.”

“Therefore, the highest efficiency and power density are obtained with the FSPM motor when ignoring any other transmission part, thus making it a competitive candidate for the MagD system. Finally, given the excellent simulation results of the FSPM motor, a downscaling prototype of the FSPM motor is fabricated, …”